# Diverse Roles of the Exon Junction Complex Factors in the Cell Cycle, Cancer, and Neurodevelopmental Disorders-Potential for Therapeutic Targeting

**DOI:** 10.3390/ijms231810375

**Published:** 2022-09-08

**Authors:** Hannah Martin, Julian Rupkey, Shravan Asthana, Joy Yoon, Shray Patel, Jennifer Mott, Zifei Pei, Yingwei Mao

**Affiliations:** 1Department of Biology, Pennsylvania State University, University Park, State College, PA 16802, USA; 2Feinberg School of Medicine, Northwestern University, 303 East Superior Street, Chicago, IL 60611, USA

**Keywords:** EJC, NMD, RBM8A, MAGOH, eIF4A3, MLN51

## Abstract

The exon junction complex (EJC) plays a crucial role in regulating gene expression at the levels of alternative splicing, translation, mRNA localization, and nonsense-mediated decay (NMD). The EJC is comprised of three core proteins: RNA-binding motif 8A (RBM8A), Mago homolog (MAGOH), eukaryotic initiation factor 4A3 (eIF4A3), and a peripheral EJC factor, metastatic lymph node 51 (MLN51), in addition to other peripheral factors whose structural integration is activity-dependent. The physiological and mechanistic roles of the EJC in contribution to molecular, cellular, and organismal level function continue to be explored for potential insights into genetic or pathological dysfunction. The EJC’s specific role in the cell cycle and its implications in cancer and neurodevelopmental disorders prompt enhanced investigation of the EJC as a potential target for these diseases. In this review, we highlight the current understanding of the EJC’s position in the cell cycle, its relation to cancer and developmental diseases, and potential avenues for therapeutic targeting.

## 1. Introduction

Processing of mRNA is crucially regulated in the mammalian cells via differential alternative splicing, translation, mRNA localization, and NMD [1]. The EJC has emerged as a significant component and regulator of mRNA processing [2,3,4]. The EJC is composed of three core components and a peripheral factor, although the auxiliary factors are integrated dependent on physiological EJC function or cell context as shown in Table 1 [5]. EJC assembly (Figure 1) follows a sequential order and is dependent on splicing activity [6,7]. Spliceosome protein and EJC core component eIF4A3 binding partner CWC22 provide this direct link between EJC deposition and splicing activity [8]. Due to their tight adherence to mRNAs after binding, EJCs participate in the majority of an mRNA’s lifecycle [9]. Therefore, the EJC’s influence on mRNA metabolism places it in an important regulatory position over various gene expression stages.

NMD, a highly conserved pathway of RNA surveillance and degradation, pinpoints abnormal mRNAs according to the presence of a premature termination codon (PTC) or other termination signatures [22]. This mechanism ensures that faulty mRNAs are not fully translated into nonfunctional proteins. This mechanism relies on EJC to help identify mRNAs with PTCs. NMD is an mRNA post-transcriptional quality control mechanism affecting about 10–25% of the mammalian genome because of its regulatory function in governing the fate of alternative splicing products [23]. As the mRNA is being translated, the ribosome senses EJCs downstream from the stop codon. Many genes have no exons after the stop codon. When the ribosome reaches a stop codon, it should not sense any more EJCs downstream. In the presence of a PTC, when the ribosome reaches the stop codon, EJCs will remain downstream of the ribosome. This signals to the ribosome that the stop codon is premature and the ribosome then recruits the SMG-1-Upf1-eRF1-eRF3 (SURF) complex to interact with the EJC to tag the mRNA via UPF1 mediated phosphorylation for degradation [24,25,26,27,28].

NMD also regulates a subset of RNAs that display special characteristics and lack PTCs. These characteristics include a long 3′UTR, an exon-exon junction >50 nucleotides downstream of a stop codon, a long upstream open reading frame (ORF), and introns in the 3′ UTR. RNAs with introns in the 3′UTR usually introduce a stop codon, or a frame-shift, which leads to a PTC downstream, causing the RNA to be tagged for degradation. Typically, NMD is identified when the ribosome reaches a stop codon and there is still an EJC downstream. In the case of an intron in the 3′UTR, this occurs when in some cases not all the exons are translated. This leaves both exon(s) and intron(s) downstream, which can also trigger NMD [25,29,30,31]. The physiological role of NMD and its corresponding dysfunction being implicated in genetic disease and cancer have resulted in tremendous interest in targeting the pathway [32,33,34,35]. Investigations of NMD pathway mechanisms indicate both EJC-dependent and EJC-independent branches that occur across differential cofactor requirements [16,36]. The EJC peripheral components involved in NMD are summarized in Table 1. The detailed mechanism of EJC-dependent NMD, associated branch pathways, and co-functional steps is complex and out of the scope of this review; although these details have been summarized further in other reviews [2,37]. 

In this review, we aim to discuss the EJC’s increasingly vital roles in the cell cycle and its corresponding implications on cancers and genetic disorders. Accordingly, we suggest NMD inhibition at both EJC-dependent and independent branches to be a viable target for further investigation for therapeutic potential.

## 2. Core EJC Components Influence Normal Cell Cycle Progression

The cell cycle is a fine-tuned process with little leeway for errors. Dysregulation in proteins that control this process can result in pathologies like cancer. Recent studies have examined the roles of EJC core proteins in the cell cycle and cancer. Mazloomian et al. showed that eIF4A3 inhibition in mammalian HCT116 and HeLa cell lines causes the G2/M checkpoint arrest, chromosome segregation abnormalities, and increased apoptosis (Figure 2) [38]. These results indicate normal eIF4A3 expression is necessary for normal cell cycle progression. Mechanistically, eIF4A3 is phosphorylated in a cell cycle-dependent manner by CDK1 and CDK2 [39]. Phosphorylation of eIF4A3 on Threonine 163 residue during the M phase inhibits the function of NMD as phosphorylated eIF4A3 decreases binding affinity to mRNA transcripts (Figure 2). Particularly, phosphorylation of eIF4A3 prevents the formation of a complete EJC complex with other EJC factors, but prompts CWC22 binding, which facilitates eIF4A3 back to the nucleus to form an active spliceosome. Dephosphorylation of eIF4A3 via an unknown phosphatase influences the M/G1 checkpoint. Overexpression of eIF4A3 is associated with poor prognosis of patients with various cancers and systematic analysis of co-expression reveals an association between eIF4A3 and known proteins involved in the cell cycle and apoptosis [39]. Interestingly, the researchers determine that the tumor necrosis factor-α (TNF-α) pathway is specifically impacted by *eIF4A3* mutations (Table 2) [40].

A study of RBM8A knockdown in HeLa cells demonstrates that genes encoding for splicing in mitosis and genome stability are dependent on the EJC complex [41]. Knockdown of RBM8A decreases MAGOH expression but not eIF4A3 expression [41]. RBM8A knockdown alters p53 splicing and causes the G2/M arrest (Figure 2) [41]. Multinucleated cells with abnormal nuclear structure and overall genome instability emerging under RBM8A knockdown support that RBM8A is required for normal cell cycle function [41]. Furthermore, another study found the cell cycle stalling at the G2/M checkpoint in human tumor cells deficient for RBM8A [42]. Specifically, centrosome maturation is negatively impacted, preventing the cells from going from mitosis to G1 [42]. The cells at M-phase subsequently undergo apoptosis via caspase activation. A properly assembled centrosome is a necessity for mitosis as its associated spindle fibers function to separate sister chromatids. A recent study has found that mRNAs are colocalized at the centrosome in a cell cycle-dependent manner [43]. The high-throughput single-molecule fluorescence in situ hybridization (smFISH) detected that eight mRNAs behave in this way: *ASPM*, *BICD2*, *CCDC88C*, *CEP350*, *HMMR*, *NIN*, *NUMA1*, and *PCNT* [43]. The centrosome-mRNA colocalization requires nascent protein translation as puromycin but not cycloheximide prevents the centrosomal mRNA localization. The polysomes with mRNA transcripts are actively transported towards centrosomes through microtubule during prophase and prometaphase [43]. Therefore, improper localization of mRNA to a mature centrosome could stand as one of the possible downstream effects of abnormal RBM8A levels in cells.

Multiple studies have investigated the effects of MAGOH deficiency in the context of neurodevelopment. Mitosis is essential in neurodevelopment in the context of cell proliferation and differentiation. In the context of melanoblast development, this process is associated with normal pigmentation of dorsal and ventral regions in adult mice (Table 2) [44]. *Magoh* depletion, namely haploinsufficiency, inhibits melanoblast proliferation [44]. While the cells are not undergoing apoptosis, the mitosis is arrested at the G2/M phase, preventing cell proliferation [44]. Another study also found that low levels of MAGOH in interneuron progenitors result in mitotic staling [45]. Consequently, this reduces the critical number of cortical interneurons. The interneuron progenitors display upregulated p53 signaling, aligning with observed apoptotic divisions [45]. *Magoh* deficiency in radial glial cell progenitors results in apoptosis, via p53 activation, upon mitosis delay [46]. In this study, changes to the microtubules that compose the mitotic spindle are evident. In addition to MAGOH, RBM8A expression regulates neurodevelopment. Consistent with the essential roles of EJCs in neurodevelopment, in a study of neural progenitor cells (NPCs), RBM8A is deemed necessary for cell cycle regulation, NPCs and interneuron progenitor proliferation [47,48]. Elevated levels of RBM8A inhibit cell cycle exit, leading to more proliferation, whereas low levels were associated with cell cycle exit and subsequent differentiation [47].

## 3. The EJC Is Implicated in a Diverse Set of Cancers

Recently, the major core and peripheral components of the EJC have been implicated in various carcinogenic pathways. Several forms of cancers have been associated with the EJC as shown in Table 3. These associations vary greatly in both the experimental paradigms utilized to establish these associations and in the exact role the implicated EJC component plays. Additionally, mechanistic insight thus far has demonstrated several pathways differentially affected such as NOTCH1, STAT3, mTOR, RAS, or MAPK among many other cellular signaling networks [42,49,50,51,52,53,54].

While a detailed summary of each of these associations is out of the scope of this review, we will highlight some of these EJC-associated cancer mechanisms below.

### 3.1. RBM8A Regulates Glioblastoma Progression

A recent study on the effect of *RBM8A* knockdown in glioblastoma cells shows that it inhibits glioblastoma cell proliferation and invasion. RBM8A increases cell proliferation and migration in glioblastoma tissues through the Notch/STAT3 pathway [56]. As RBM8A mediates DNA binding and regulates the downstream genes by interacting with a transcriptional factor STAT3 [58], the protein levels of Notch, phospho-STAT3, and phospho-H3 are significantly decreased by *RBM8A* knockdown [56]. The opposite effects have been detected in RBM8A overexpression that transwell analysis confirms an increase in cell migration and invasion of glioblastoma cells when RBM8A is overexpressed but a dramatic decrease when RBM8A is knocked down. To further examine *RBM8A* function *in vivo*, authors demonstrated that RBM8A knockdown leads to a reduction of Notch1 protein level in tissues and a slow growth of the tumor cells in a xenograft mouse model. Furthermore, higher expression of RBM8A correlates to a worse prognosis in patients. These findings provide a possible avenue for effective diagnosis and treatment of glioblastoma and other cancers [56,59,60].

### 3.2. RBM8A Affects Apoptosis in Human Cell Models of Adenocarcinoma

RBM8A functions as an important apoptotic modulator. Knockdown studies of *RBM8A* in human cervical carcinoma cell line (HeLa) and human lung adenocarcinoma cell line (A549) further established the role of RBM8A in cell cycle regulation. *RBM8A* knockdown inhibits tumor cell activity, arrests the cell cycle at the M phase, and promotes apoptosis [42,59,60]. The apoptotic defining characteristics featured abnormal centrosomes in cancer cells, supporting the importance of *RBM8A* in apoptosis regulation [61]. *RBM8A* depletion regulates the expression of pro-apoptotic genes, such as *Bcl-Xs* and *Bim*, and tumor development associated with mRNA splicing regulation [59,60,62]. Results also show that RBM8A expression affects tumor diameter, degree of pathological differentiation and clinical stage, overall survival, and progression-free survival. Interestingly, the knockdown of different EJC/NMD components gives different effects on *Bcl-Xs* splicing. Downregulation of RBM8A and eIF4A3 or auxiliary (RNPS1, Acinus, and SAP18) components of the EJC, promotes the production of the proapoptotic *Bcl-x_S_* splice variant. However, UAP56, Aly/Ref, and TAP or NMD factors (MNL51, Upf1, Upf2, and Upf3b), do not produce the same effect [62].

### 3.3. RBM8A and eIF4A3 Contribute to the Pathogenesis of Hepatocellular Carcinoma (HCC)

RBM8A is important in cell development, differentiation, metabolism, and regulation of the cell cycle. Elevated RBM8A expression has been observed in hepatocellular cellular carcinoma [57]. This overexpression has been linked to poor patient prognosis [59]. In HCC, overexpression of RBM8A increases cell proliferation, migration, and invasion into surrounding tissues. KEGG analysis has revealed that the following cellular functions are impacted in these patients: DNA replication, cell cycle, ribosomal activity, and spliceosomal activity [63]. In line with the role of cell cycle regulation, cell cycle-related kinases: CDK1, CDK2, and CHEK1, are impacted [63]. The transcription factor, E2F1DP1RB, which modulates mitosis at the G1/S transition, is also affected [63]. Overexpression of RBM8A upregulates the epithelial-mesenchymal transition (EMT) signaling pathway. The EMT is implicated in tumor cell metastasis and the resistance to chemotherapy treatments [57]. The activation of the EMT via RBM8A promotes HCC growth, which plays a significant role in the regulation of oxaliplatin (OXA)-based systemic chemotherapy resistance [57]. A high level of RBM8A is associated with a higher HDAC9 level, resulting in the resistance effect of OXA, a key chemotherapy drug [57]. Moreover, higher RBM8A expression was detected in advanced HCC than in early-stage HCC, suggesting a good correlation between RBM8A level and carcinogenesis [59,60]. These results indicate the possibility of RBM8A as an HCC prognostic tool.

Besides RBM8A, eIF4A3 is overexpressed in HCC tissues compared to normal tissues, which is indicative of a poor prognosis. Functional gene network analyses revealed a connection between eIF4A3 and various chemokine pathways, cell cycle and spliceosome pathways, several cell cycle regulatory kinases, and tumor-associated transcription factors [38]. Through antagonistic binding to WDR66 with *miR-2113*, eIF4A3 promotes HCC cell proliferation and EMT function. Downregulation of eIF4A3 leads to G2/M phase arrest, increased apoptosis, as well as abnormalities in centrosome-spindle formation [38]. Moreover, eIF4A3 has been associated with RNA stress granule formation and maintenance, indicating another possible mechanistic link between the EJC and cancer [38]. Together, these findings provide another possible mechanism for therapeutic development through manipulating the EJC level.

## 4. The EJC Plays a Crucial Role in Neurodevelopmental Disorders

The EJC components have been studied widely for their role in gene regulation, as well as their significant effects on downstream regulation of cell cycle and splicing. In addition to cancer, abnormal expression of RBM8A plays a role in thrombocytopenia and neurodevelopmental diseases, such as autism and schizophrenia. However, the pathologic mechanisms behind the role of *RBM8A* in these diseases are not yet fully understood.

### 4.1. The EJC Controls Cell Division of Neural Stem Cells (NSCs)

The development of the nervous system requires a balance between the proliferation and maintenance of a pool of NSCs and the differentiation/migration of NSCs to the cortical and subcortical regions of the brain. Cortical development begins with the generation of radial glial cells around embryonic day 9/10 (E9/10) in mice. NSCs undergo symmetric division or asymmetric division to ensure that the stem cell pool is maintained, while enough cells differentiate into neurons and glial in the brain. In the development of the cortex, neurogenesis relies on asymmetric division, which produces a stem cell and a cell that will terminally differentiate. This is in contrast to symmetric mitotic division, which results in two identical daughter cells. Asymmetric division is regulated by the orientation of the mitotic spindles that determine the position and identity of the daughter cells [64,65,66]. During asymmetric division, cleavage planes are usually within 30° of the apicobasal axis. This orientation is consistent through both proliferative and neurogenic phases [67].

Previous studies have shown that EJC components are highly associated with neurogenesis. *Magoh* haploinsufficiency is a prime example of how changes in the orientation of mitotic spindles can result in deficits in the asymmetric division. *Magoh* haploinsufficiency results in microcephaly and leads to increased cell division along the horizontal plane [68]. Cells that divide horizontally, are destined to become basal daughter cells and post-mitotic neurons. Thus, *Magoh* haploinsufficiency significantly prolongs mitosis of NSCs and leads to increased ectopic neuron differentiation [46,68]. Consistently, haploinsufficiency for each core EJC component causes microcephaly, depletion of NSCs, and extensive apoptosis in both neurons and NSCs in mice [69]. The microcephaly phenotypes are significantly more severe in *Rbm8a* and *Eif4a3* mutant mice, with an average reduction of 70%, compared to *Magoh* in postnatal brains. However, this is thought to be due to a second homolog of *Magoh.* Together, NSCs and neurogenesis analyses in embryonic brains demonstrate that deletions in all three EJCs contribute to NSC division defects and induction of apoptosis to a similar extent, leading to microcephaly [69].

Three EJC genes are closely related in pathophysiology as each mutant alters each other’s overlapping transcription levels. Upregulation of p53 transcript observed in all three EJC mutants suggests that p53 activation may be involved in pathogenesis. Combined with a previous finding in clinical studies that *RBM8A* deletion is closely related to human microcephaly, attenuation of p53 in *Rbm8a* mutants shows a partial but significant rescue of microcephaly and apoptosis [70,71]. This suggests that p53 activation plays a role in neuronal loss, thus leading to microcephaly. However, it is unclear whether p53 activation leads to induced apoptosis or delayed mitosis. Further experiments are needed to assess this relationship.

### 4.2. EJC Modulates RNA Splicing during Development

The EJC factors assembly at 20–24 nucleotides upstream of mRNA exon-exon junctions stepwise in an orderly pattern during RNA splicing [72], and provide a critical link between pre-mRNA splicing and subsequent events including mRNA export, localization, translation, and NMD [73]. Clearly, RNA splicing modulates gene expression and plays a critical role in various developmental stages, particularly in the nervous system [74].

EJC factors can regulate mRNA splicing of key genes in multiple essential signal pathways to modulate development. In *Drosophila*, *Magoh* mutations cause specific defects in photoreceptor differentiation due to altered MAPK splicing [75]. Interestingly, other EJC factors like RBM8A and eIF4A3 are also required for MAPK splicing, supporting a nuclear role of EJC in the splicing of a subset of transcripts [75]. Another splicing substrate of EJC in *Drosophila* is *dlg1*, a cell polarity gene [76]. Dlg1 protein promotes Dvl protein stabilization and stimulates the Wnt signaling pathway. Thus, EJC can modulate the activation of the Wnt pathway via Dlg1 splicing [76]. A genome-wide approach, isolation of protein complexes and associated RNA targets (ipaRt), has been used to identify EJC-bound RNAs that are enriched with genes regulating differentiation and development in adult *Drosophila* [77]. Moreover, EJC binds to strong splicing sites, CG-rich hexamers [77]. In addition to large introns, EJC can also regulate the splicing of many mini-introns in *P. tetraurelia*, suggesting that a fine-tuning expression of EJC genes is required for steady intron removal [78].

Core EJC factors are essential for neurodevelopment [69]. Deletion of each core EJC factor leads to microcephaly in mice. Consistent with the notion, spliceosome dysregulation is implicated in RNAseq analyses for all EJC mutant mice [69]. Alternative pre-mRNA splicing is normally coupled with NMD to selectively degrade some splicing isoforms while permitting the expression of other isoforms. This mechanism induces developmental expression of the synaptic scaffold protein PSD-95 in mammals but not in invertebrates, suggesting an evolutionary mechanism regulating neural-specific expression of PSD-95 [79]. Interestingly, exon 5 splicing of a proapoptotic gene *Bak1* is subject to developmental regulation [80]. In NPCs, splicing regulator PTBP1 is high and promotes exon 5 skipping of *Bak1*, which allows *Bak1* to escape from NMD and produce functional BAK1 protein for activating apoptosis. Upon differentiation into neurons, PTBP1 level goes down, which leads to *Bak1* exon 5 inclusion and trigger NMD mechanism to break down pro-apoptotic *B**ak1* transcripts and enable neuronal survival. Consistently, germline deletion of *Bak1* exon 5 increases neuronal apoptosis and leads to postnatal mortality [80], supporting a critical function of splicing and NMD in controlling brain development.

### 4.3. RBM8A Contributes to Thrombocytopenia and Absent Radius (TAR) Syndrome and Neurodevelopmental Disorders

The proximal region of 1q21 copy number variation (CNV) is often referred to as the TAR region, as a microdeletion of this region has been found in patients with the TAR syndrome in previous clinical studies [63,71,72]. Heterozygosity of *Rbm8a* deletion resulted in platelet deficiency and increased tail bleeding in knockout mice [81]. Further immunohistochemical staining (IHC) demonstrated that *Rbm8a* knockout reduces the average size of megakaryocytes and impairs its differentiation as it forms clusters in the spleen, resulting in splenomegaly in mice [81]. Flow cytometry of human erythroleukemia (HEL) cells stained with megakaryocyte markers shows that *RBM8A* depletion causes cell-cycle arrest and attenuates platelet differentiation. Consistent with the finding of an inverse relationship between RBM8A expression and p53 protein levels in HEL cells, p53 inhibitor and p53 knockout not only increase the transcriptional levels of *Rbm8a* but also partially rescue both megakaryocyte differentiation and platelet counts [81].

A dosage effect of RBM8A has been found in many clinical manifestations, followed by altered cell proliferation and differentiation. Overexpression of RBM8A promotes embryonic NSC proliferation and suppresses neuronal differentiation, whereas the knockdown of *Rbm8a* has the opposite effect [46,47]. Thus, abnormal expression of RBM8A has been studied in neurodegenerative, neurodevelopmental, and neuropsychiatric research regarding Alzheimer’s disease, autism spectrum disorder, and schizophrenia [82]. *Rbm8a* deficits lead to prolonged mitosis in NSCs and intermediate progenitors, as well as the downregulation of Lis1 during neurogenesis together with MAGOH heterodimers [69]. The unstable MAGOH heterodimer complex is highly associated with microcephaly due to apoptosis.

## 5. Targeting the EJC and NMD in Diseases

In 2001, it was proposed to utilize drugs or siRNA to inhibit NMD for cancer treatment [60,83,84]. The mRNA stability has been associated with the prevalence of malignant tumors, which supports the need to study NMD in different cancers [60]. Three cellular pathways, Ras/MAPK, JAK/STAT3 and NF-κB, have been associated with RBM8A’s role in liver cancer [59,60]. Depletion of RBM8A inhibits all three pathways. MAPK protein synthesis is declined, thereby inhibiting the Ras/MAPK pathway [60]. RBM8A depletion also inhibits the JAK/STAT3 pathway and decreases STAT3 DNA-binding activity [60]. The effects of RBM8A on these signaling pathways suggest a critical role of RBM8A in tumorigenesis. Increasing NMD efficiency has recently been proposed to target carcinogenesis [60,85]. Quantification of NMD-triggering versus NMD-evading PTC leads to the classification of tumor-suppressor genes [85].

As previously found that *Eif4a3* knockdown induces apoptosis in cancer cells, allosteric inhibitors have been investigated as a potential NMD suppression tool [86]. Frameshift insertion/deletions (fs-indels) are normally targeted and degraded by NMD. However, some fs-indels are in the genomic regions that can escape NMD. Studies of T cell reactivity have shown that antigens derived from fs-indels, specifically those with elongated ORFs, can elicit an immune response [87]. These NMD-escape mutations could be possible targets for immunotherapy because these mutations can stimulate anti-tumor response through checkpoint inhibitors (CPI) with clear clinical benefits [87].

Several other conserved regulatory pathways can modulate NMD efficacy. First, *miR-128* inhibits NMD by targeting UPF1 and CASC3 during neuronal differentiation [88]. Second, the cytoskeleton is responsible for the transportation of mRNAs carrying PTCs and ribonucleoproteins (RNPs) to the processing body (P-body) for degradation. Disruptions of different cytoskeletons exert different effects. Interruption of actin function by polymerization inhibitors blocks NMD and activates PTC readthrough, whereas microtubule disruption only inhibits NMD. PTC readthrough requires UPF proteins’ presence in P-bodies, suggesting that different cytoskeletons involve in various NMD stages [89]. Third, mRNA itself contains a wide variety of features to escape NMD. The eukaryotic release factor 3a (eRF3a) close to an upstream termination codon inhibits NMD in a manner that is poly-A binding protein C1 (PABPC1) dependent, while eIF4G does so in a PABPC1-independent manner [90]. A cis-element positioned downstream of the termination codon in the 3′ UTR has been identified to inhibit NMD and is proposed to do so through efficient translation termination. The same study identifies another cis-element in the 3′ UTR that inhibits NMD through a different but unclear mechanism [91]. Fourth, multiple RNPs regulate NMD efficacy. Heterogeneous nuclear ribonucleoprotein L (hnRNP-L) protects oncogenic BCL2 mRNA from NMD. HnRNP-L shares structural similarities with polypyrimidine tract binding protein 1 (PTBP1), which is also known to protect mRNA from NMD. Both hnRNP-L and PTBP1 aid in NMD evasion by attenuating UPF1 binding to mRNA [92]. Poly-A binding protein 1 (PABPC1) and other peptides containing the PABP-interacting motif 2 (PAM2) have been shown to inhibit NMD degradation by binding to 3′ UTRs [93]. FUS mutations in patients with amyotrophic lateral sclerosis (ALS) cause protein inclusion and sequester translation, thereby affecting NMD. In cells derived from ALS patients, UPF3A is decreased but UPF1 and UPF3B are increased [94]. Thus, abnormal NMD function could contribute to neurodegenerative diseases like ALS.

Several NMD inhibitors have been discovered or synthesized to explore the therapeutic potential of NMD inhibition [95,96,97,98]. A high-throughput screen that monitors the RNA-dependent ATPase activity of the eIF4A3−MLN51 complex, has identified 1,4-Diacylpiperazines as a selective NMD inhibitor [86] and the authors further optimize the prototype compound into several highly-selective noncompetitive inhibitors of eIF4A3 [99]. Antisense oligonucleotides have been generated to reduce the expression of different NMD factors. Knockdown of *Upf1*, *Upf2*, *Smg1*, and *Smg6* suppresses NMD, and downregulation of *Upf3b*, *Smg5*, and *Smg7* moderately affect NMD. However, *Smg8* and *Smg9* knockdown demonstrates no significant effect [100]. *Upf3b* inhibition minimally affects the overall transcriptome while stabilizing both PTC-containing dystrophin mRNA and coagulation factor IX mRNA in mouse models [100].

Unbiased high-throughput chemical screens using a multi-colored bioluminescence NMD reporter system and FDA-approved drugs have revealed some interesting hits, including a group of cardiac glycosides, including ouabain and digoxin, as potent inhibitors of NMD [101]. Cardiac glycosides target Na-K ATPase on the cell membrane to increase intracellular Ca2^+^ levels and inhibit NMD [101]. Through profiling multiple endogenous NMD targets, Zhao et al. identified several NMD modulators, including the anticancer drug homoharringtonine [102]. Given the diversity of methods that can be used to suppress NMD, NMD inhibition holds a promising therapeutic potential for ameliorating a variety of disease phenotypes.

## 6. Conclusions

The scientific literature clearly evinces the powerful influence the EJC exerts over many cellular processes. Alongside the NMD pathway, the EJC regulates the cell cycle and therefore may be associated with several cell cycle dysfunctions, carcinogenic pathways, and neurodevelopmental and neurodegenerative diseases. NMD emerges as a target of inhibition in both an EJC-dependent and independent manner for future therapeutic potential. Although progress has been made in the discovery of the roles of the EJC and NMD in cell cycle dysfunction and associated pathologies, further research is still necessary to reveal key mechanistic links within the cell cycle in normal developmental and pathological conditions.

## Figures and Tables

**Figure 1 ijms-23-10375-f001:**
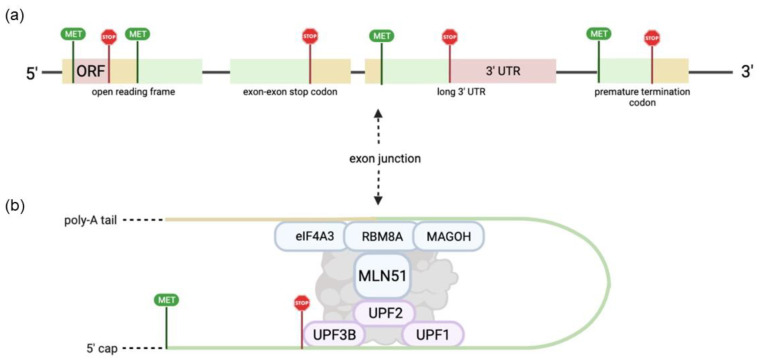
Graphical representation of EJC binding in mRNA during NMD. (**a**) NMD features that often trigger mRNA degradation, include upstream open reading frames, exon-exon stop codons, a long 3′ UTR, and a premature termination codon. (**b**) NMD cofactors UPF3B, UPF2, and UPF1 bind upstream of the exon junction, triggering exon-junction complex binding and NMD. Only core EJC factors are shown.

**Figure 2 ijms-23-10375-f002:**
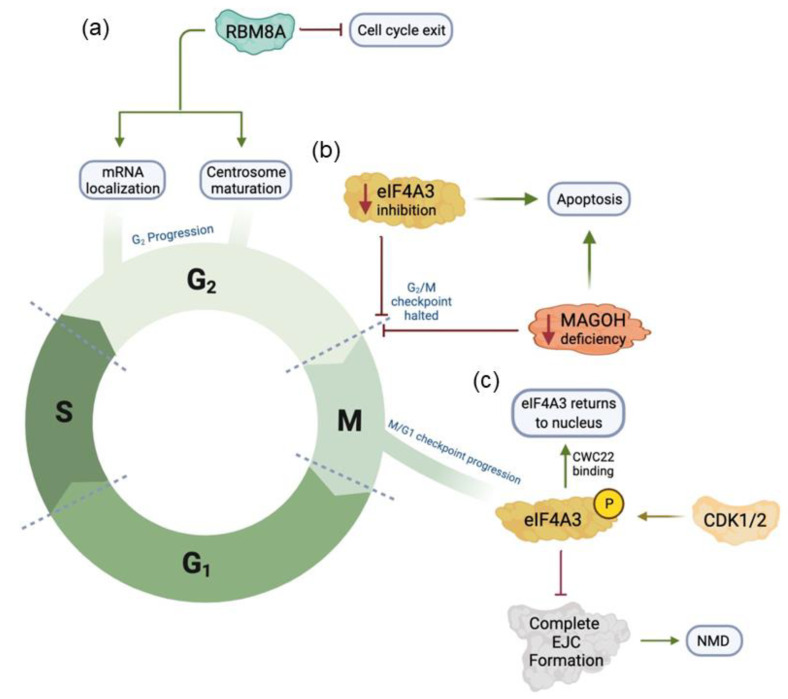
Graphical representation of the function of EJC components in the cell cycle. (**a**) RBM8A inhibits cell cycle exit and is essential for progression past the G_2_/M checkpoint through mRNA localization and/or centrosome formation. (**b**) Inhibition of eIF4A3 has been shown to cause apoptosis at the G_2_/M checkpoint. MAGOH deficiency in interneuron and glial progenitors has been shown to cause apoptosis through p53 activation. (**c**) Phosphorylation eIF4A3 by CDK1/2 results in NMD inhibition and progression past the M/G_1_ boundary.

**Table 1 ijms-23-10375-t001:** EJC core and major peripheral components associated with functional roles in context of various EJC functions. Included are key references in which EJC component structural integration is detailed.

EJC and NMD Factors	Functional Role	References
ACIN1	Alternative splicing	[10,11]
ALYREF	mRNA localization	[12]
CWC22	Alternative splicing	[8,13]
eIF4A3	Invariable EJC core component ubiquitous in all EJC functions	[8,13]
IPO13	mRNA localization	[14]
MAGOH	Invariable EJC core component ubiquitous in all EJC functions	[15]
MLN51	Alternative splicing, mRNA localization, mRNA translation, and NMD	[10,16]
PNN	Alternative Splicing	[10,11]
PYM	mRNA translation and EJC disassembly	[17]
RBM8A	Invariable EJC core component ubiquitous in all EJC functions	[18]
RNPS1	Alternative splicing and mRNA translation	[19]
SAP18	Alternative Splicing	[10,11]
SMG6	NMD	[20]
UPF1	NMD	[20]
UPF2	NMD	[20]
UPF3A	mRNA translation and NMD	[21]
UPF3B	mRNA translation and NMD	[21]

**Table 2 ijms-23-10375-t002:** Summary of cell cycle abnormalities caused by dysfunction of core EJC components. Included are key references in which EJC cell cycle abnormality is detailed.

EJC Component	Dysfunction	References
eIF4A3	Deficiency results in G2/M cell cycle arrest, chromosome segregation abnormalities, and apoptosis. Mutations of eIF4A3 specifically impact Tumor Factor-α pathway.	[38,40]
RBM8A	Deficiency results in G2/M cell cycle arrest, multinucleated cells with abnormal nuclear structure, and genome instability. Improper mRNA localization to mature centrosome has also been observed.	[41,42,43]
MAGOH	Deficiency inhibits melanoblast proliferation via arrest of cell cycle at G2/M and stalled mitosis in interneuron progenitors. Also results in apoptosis of radial glial cell progenitors.	[44,45,46]

**Table 3 ijms-23-10375-t003:** EJC-associated cancers with implicated EJC components and references.

EJC Associated Cancers	EJC Component(s) Implicated	References
Breast Cancer	MLN51, eIF4A3	[51,52]
Cervical cancer	RBM8A	[49]
Colorectal Cancer	eIF4A3	[53]
Gastric Cancer	MAGOH	[55]
Glioblastoma	RBM8A	[56]
Hepatocellular carcinoma	RBM8A, MAGOH, eIF4A3	[38,57]
Lung Adenocarcinoma	RBM8A	[42,50]
Non-small-cell lung carcinoma	RBM8A	[50]
Ovarian Cancer	eIF4A3	[54]

## Data Availability

Not applicable.

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
