# Peer review of "Diverse Roles of the Exon Junction Complex Factors in the Cell Cycle, Cancer, and Neurodevelopmental Disorders-Potential for Therapeutic Targeting"

_ijms, 2022, doi:10.3390/ijms231810375_

Round 1
Reviewer 1 Report
The authors focus on the EJC and aim to highlight the EJC’s position in the cell cycle, its relation to cancer and developmental diseases, and potential avenues for therapeutic targeting. However, there are several problems that deduct the quality of this manuscript. My comments are listed below:
1. The authors mainly focus on the possible functions/effects of individual components of the EJC on cell cycle and/or cancer development. Thus, the manuscript title does not match the manuscript text. It remains unclear what’s the functional effect of EJC as a whole on cell cycle and/or cancer development. For example, the functions of RBM8A discussed in this review may be different from its role when acting as a component of the EJC.
2. In sections 2-4, the authors did not discuss the effect of the alteration of individual components on the EJC itself.
3. In section 5, how does targeting of individual components impact the EJC and how the affected EJC alter cell cycle / NMD?
Author Response
We thank for reviewer 1’s comments.
- The authors mainly focus on the possible functions/effects of individual components of the EJC on cell cycle and/or cancer development. Thus, the manuscript title does not match the manuscript text. It remains unclear what’s the functional effect of EJC as a whole on cell cycle and/or cancer development. For example, the functions of RBM8A discussed in this review may be different from its role when acting as a component of the EJC.
--We thank the reviewer for this suggestion. We have changed the title to reflect the diverse roles of EJC.
2. In sections 2-4, the authors did not discuss the effect of the alteration of individual components on the EJC itself.
-- We thank the reviewer for this comments. We have added the discussion into main text as high-lightened in the Tracking Changes function, if the primary literatures have the experimental results on the effect on alteration of EJC. However, many literatures only focus one EJC component on certain functions without studying the effects on the EJC as a whole complex.
3. In section 5, how does targeting of individual components impact the EJC and how the affected EJC alter cell cycle / NMD?
-- We thank the reviewer for this comments. Similarly, we have supplemented discussion.
Reviewer 2 Report
The authors in the manuscript present a review on EJC highlighting the role in the cell cycle. Though , recently a few reviews have been published in the same topic, the authors have focussed on the specific role in cell cycle.
I would recommend a couple of points:
1. I would recommend to add a few lines on how EJC is important for Splicing and NMD.
2. Involvement of EJC in neural cell division has been studied well and might warrant a small paragraph describing the various findings.
Author Response
We thank for reviewer 2’s support.
- I would recommend to add a few lines on how EJC is important for Splicing and NMD.
-- We thank the reviewer for this comments. We have significantly expanded discussion on splicing in section 3.2 and added a new section of 4.2. We have added more discussion on NMD in section 1.
- Involvement of EJC in neural cell division has been studied well and might warrant a small paragraph describing the various findings.
-- We thank the reviewer for this comments. We have added more content on NSC division in section 4.1.
Round 2
Reviewer 1 Report
The manuscript is suitable for publication now.